# A Normative Framework for Reasoning in Language Models

## Abstract

Large Language Models (LLMs) increasingly exhibit advanced abilities, enabled by techniques such as chain-of-thought prompting and test-time deliberation. However, they continue to struggle with tasks that demand complex reasoning, prompting debate over whether their outputs reflect genuine reasoning processes or merely statistical pattern generation. These difficulties stem in part from the absence of a unified framework for explaining and assessing reasoning in LLMs, which limits our ability to diagnose errors, establish bounds, and design effective interventions. In this paper, we propose a normative framework that characterizes reasoning as probabilistic inference over propositions and we show how this abstraction can be instantiated in LLMs. Within this framework, we provide a typology of reasoning modes, formalise success criteria for proposition-level correctness, and derive a taxonomy of failure modes. For each class, we map model-level requirements to LLM-level implementation constraints and identify potential remedies. Finally, we outline a roadmap for improving proposition-level accuracy under tractable approximations. Our contribution is both diagnostic and prescriptive: an account of what it means for LLMs to reason, where and why current systems fail, and how to close the gap.

## 1 Introduction

Reasoning is a defining feature of human cognition and a long-standing aspiration for artificial intelligence (Pearl et al.; Fagin et al., 2004; Russell & Norvig, 2022; Ying et al., 2025). It is typically understood as the process of drawing conclusions from premises or evidence and assessing the validity or strength of the connections between steps (Pearl, 2014; Russell & Norvig, 2022). It spans a variety of areas, from logical and mathematical reasoning to common sense reasoning (Pearl, 2014; Russell & Norvig, 2022).

Large language models (LLMs) now display reasoning-like behaviour: they can generate multi-step solutions, follow chains of thought, and solve tasks that appear to demand structured inference (Graves, 2016; Cobbe et al., 2021b; Wei et al., 2022b; Besta et al., 2025; Wu et al., 2024; Chen et al., 2025a; Zhu et al., 2025). Modern reasoning language models, with enhanced test-time compute capabilities, have demonstrated increased reasoning skills, yielding state-of-the-art results on benchmarks (DeepSeek-AI et al., 2025; Chen et al., 2025a; Besta et al., 2025). Yet these models exhibit clear limitations, often failing on seemingly simple reasoning tasks, both due to shortcomings in the reasoning process itself and in their ability to represent the underlying structure of beliefs (Xu et al., 2024; Wu et al., 2024). Performance gaps have been documented in various domains: formal logic (Parmar et al., 2024; Liu et al., 2025), causal and counterfactual reasoning (Wang, 2024; Chen et al., 2025b), inductive generalization (Luo et al., 2023), mathematical reasoning (Cobbe et al., 2021a; Ahn et al., 2024; Yan et al., 2024; Marjieh et al., 2025), long-horizon reasoning (Ling et al., 2025), and mixtures of reasoning tasks (Chollet et al., 2025; Phan et al., 2025). A common thread across these failures is misrepresentation of reasoning: LLMs often produce surface-level approximations rather than correct intermediate steps, which can prevent them from generalizing beyond the distribution of their training data (Bender et al., 2021). This raises the fundamental question of the extent to which modern language models are capable of genuine reasoning. Answering this is made more difficult by the absence of a systematic framework for characterizing reasoning and for precisely distinguishing successful from unsuccessful reasoning processes (Wu et al., 2024; Chen et al., 2025a; Zhu et al., 2025).

In this paper, we address this gap by proposing a unifying framework for reasoning. This framework serves as a normative framework for understanding and evaluating reasoning capacities in language models. Our contributions are fourfold. First, we present a formal framework that provides a unified normative account of reasoning across tasks. Second, we analyze how LLMs approximate reasoning processes within their current architectures and training regimes. Third, we identify a taxonomy of potential failures of LLM reasoning and propose improvements. Finally, we examine the broader challenges of implementing these solutions in LLMs and devise a roadmap that highlights both current obstacles and potential paths forward.

## 2 A Normative Framework of Reasoning

Reasoning operates on propositions (Enderton, 2001; Jaynes, 2003; Koller & Friedman, 2009), where a proposition is a statement that can, in principle, be true or false, depending on the state of the world. In classical logic, reasoning operates on propositions with absolute truth values (Enderton, 2001). Probabilistic reasoning extends this framework to uncertainty over propositions (Koller & Friedman, 2009;?). The formal framework for probabilistic reasoning then derives from the foundations of probability theory (Kolmogorov). Specifically, a probability space is given by a triple $(\Omega, F, P)$, where $\Omega$ is the sample space of all possible states of the world, $F$ is the event space, and $P$ is a probability measure that assigns each event $A \in F$ a degree of belief $P(A)$ between 0 and 1, with $P(\Omega) = 1$. In this setting, a proposition corresponds to an event $A \in F$, the set of all states of the world in which the proposition holds. The probability $P(A)$ quantifies the degree of belief that the proposition is true and the joint distribution $P$ encodes all possible assignments of truth values. Reasoning can then be construed as inference within this probability space (Kolmogorov; Jaynes, 2003).

While probability theory defines how beliefs are represented, it does not specify how they should change when new information arrives. To connect prior and posterior beliefs, we require an updating principle, that is, a rule that specifies how probabilities should change in light of new evidence. While various updating rules have been proposed, the one that guarantees coherence of belief is Bayes' rule (Cox, 1946). It states that, upon learning of an event $A$, the probability of a hypothesis $h$ should be updated to $P_{\text{new}}(h) = \frac{P(A|h)P(h)}{P(A)}$. Baye's rule thus provides the bridge from new evidence to a revised probability distribution, yielding beliefs that more adequately account for the observed data. Deductive inference, for instance, appears as the special case in which probabilities reduce to 0 or 1, which produces classical logical inference. This framework can be extended to causal models by introducing the $do()$ causal operator that isolates specific causal paths on the Bayesian network, thereby enabling reasoning about interventions and counterfactuals (Pearl, 2009a;b).

At a large enough scale, the full joint probability distribution over all propositions quickly becomes intractable. A natural solution is to factorize this space by using independence and conditional independence relations and Bayesian networks provide such a data structure (Koller & Friedman, 2009; Pearl, 2014). This type of network simplifies the full joint distribution by factorizing over conditional independence. Introducing latent variables can make this factorization even sparser without loss of information (e.g., by explaining correlations with hidden causes). Even so, the factorized network can remain large. One can also introduce lossy compression by imposing approximate independencies or by aggregating variables within the network; in other terms, by relaxing the Markov boundary. Such choices trade fidelity for tractability: larger networks preserve more of the true joint distribution but at greater computational cost (Pearl, 2014). Different compression strategies thus reflect modeling choices about how closely to approximate the underlying distribution, and in some cases correct answers can still be recovered even at high levels of compression.

Such networks support inference over propositions: once a network specifies a joint distribution, conditioning on evidence $E$ yields a unique posterior $P_E(Q) = P(Q \mid E)$ for any query $Q$. Bayesian updating serves as the normative rule for belief revision: it is the unique coherence-preserving way to update degrees of belief (Bovens & Hartmann, 2004; Pearl, 2014). Reasoning in a Bayesian network can proceed in different directions on the graph and can be characterized by distinct modes, summarized in Table 1.

Table 1: The four main modes of reasoning and their realisation in Bayesian networks.

| Type | Explanation | Illustration in a BN |
|---|---|---|
| Deductive | From premises to conclusions. | If $P(Y \mid X) = 1$ and $X$ is true, then $Y$ must be true. |
| Inductive | Generalise from observations to rules or parameters. | Estimate $\theta$ from data $D$: $P(\theta \mid D) \propto P(D \mid \theta)P(\theta)$. |
| Abductive | Infer the most plausible cause of observations. | Given effect $E$, choose the hypothesis $H$ that maximises $P(H \mid E)$. |
| Causal | Predict effects of interventions. | Intervention: $P(Y \mid do(X = x))$, computed by modifying the graph to fix $X = x$. |

## 3 Reasoning in Language Models

Modern language models adopt an autoregressive architecture: at each step they generate a probability distribution over the token vocabulary conditioned on all previously generated tokens (Vaswani et al., 2017; Zhao et al., 2023; Minaee et al., 2024; Shen et al., 2023). The vocabulary $\Sigma$ functions as an alphabet, and any finite sequence of tokens from $\Sigma$ constitutes a string in the usual formal-language sense. The set of all such strings is the Kleene star $\Sigma^*$ (Cotterell et al., 2023). By iterating the conditional next-token distribution $N$ times, an autoregressive model defines a probability distribution over strings in $\Sigma^*$ (Huang & Chang, 2023; Sun et al., 2024). Since the model can, in principle, generate sequences of unbounded length, this induces a distribution over a countably infinite set of possible strings, each corresponding to a distinct trajectory of the generative process.

Within $\Sigma^*$, only a subset corresponds to propositions, that is, strings that express truth-evaluable statements (e.g., "The cat is on the mat"). Let $\Sigma^*_{\mathrm{prop}}$ denote the set of all such propositional strings; by definition $\Sigma^*_{\mathrm{prop}} \subseteq \Sigma^*$. Reasoning, insofar as it concerns the preservation of truth values, operates within this propositional subset (as discussed in Section 2). Logical inference, for instance, only applies only to elements of $\Sigma^*_{\mathrm{prop}}$, since the rules of valid reasoning presuppose that the expressions involved admit truth values. Given that an autoregressive model defines a probability distribution over $\Sigma^*$, it also induces a distribution over the propositional subset $\Sigma^*_{\mathrm{prop}}$. The probability of a proposition is then given by the total probability mass the model assigns to the set of strings that express that proposition, relative to a given context or prompt.

In this way, propositional content enters the training objective only implicitly. When the model is conditioned on long token sequences, it learns the empirical distribution of those sequences in the corpus that correspond to propositional statements. Because next-token prediction is equivalent to maximizing the likelihood of complete strings in $\Sigma^*$, improvements to the model's approximation of the full sequence distribution necessarily improve its approximation of the propositional subset $\Sigma^*_{\mathrm{prop}} \subseteq \Sigma^*$. Provided that the training corpus contains a sufficiently varied sample of propositional language, the statistical relations among propositions are learned as part of this process. An autoregressive language model therefore approximates the joint distribution over all strings in $\Sigma^*$, and therefore over the propositions embedded within them.

The model internalizes regularities that link propositions expressed in text, so that the presence of one proposition can inform the likelihood of another during generation (Jiang, 2023). In other terms, the next-token prediction objective can be viewed as prediction conditioned on prior propositional content, with additional variability introduced by the token-level training objective. LLMs thus induce an approximate propositional distribution $\hat{P}(V)$ over a model-level set of propositional variables $V$ which can be represented as a Bayesian network (Wang et al., 2023a). We denote by $P^*(V^*)$ the true joint distribution over the underlying propositional variables $V^*$. A language model with parameters $\theta$ defines a token sequence distribution $P_\theta(x_{1:T})$, which induces the approximate propositional distribution $\hat{P}(V)$ through the mapping from token strings in $\Sigma^*_{\mathrm{prop}}$ to propositional variables in $V$.

The propositional space is vast, and even a factorized network cannot represent it in full. LLMs therefore rely on compression through high-dimensional latent representations (Goodfellow et al., 2016; Vaswani et al.,

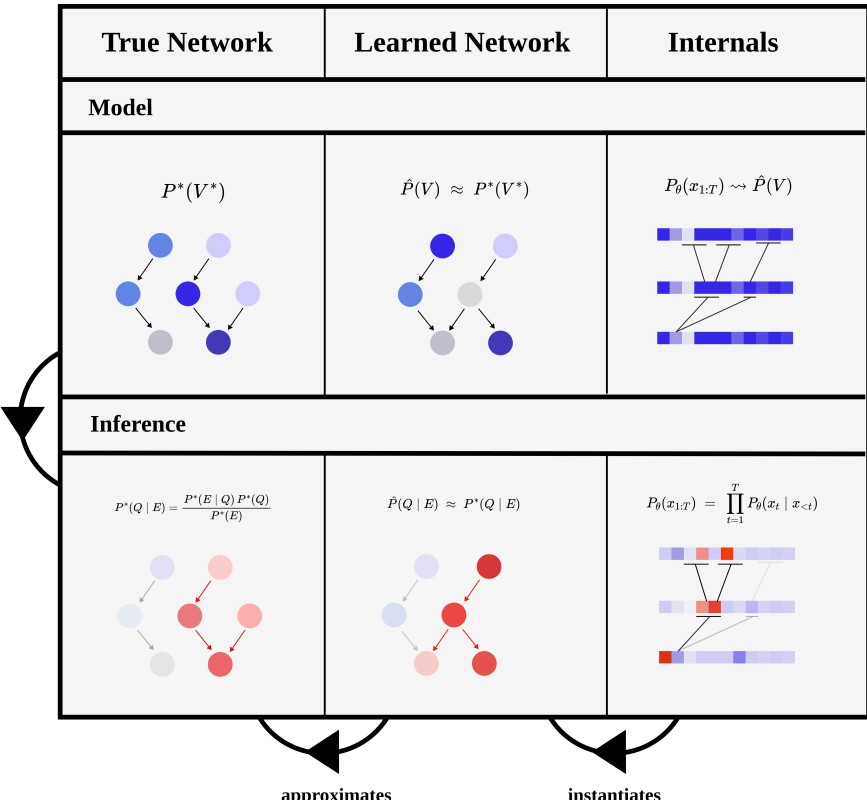

Figure 1: **Conceptual map from distributions to implementation.** Left: the true Bayesian network with joint $P^*$. Center: a learned Bayesian network $\hat{P}(V)$ approximating $P^*(V^*)$. Right: internal computations that instantiate the learned BN during inference.

2017). These latent variables act as compressed features in the representational space, enabling the model to generate sentences by traversing this compressed space (Bengio et al., 2013; Jiang, 2023; Wang et al., 2023a). Learned during training, such compressed representations constitute a first form of reasoning: they capture regularities at the proposition level. As model capacity increases, these regularities are represented with greater fidelity, which enables better approximation of the underlying propositional relationships. Correspondingly, reasoning-like abilities tend to emerge more clearly in larger models (Wei et al., 2022a; Jiang, 2023).

This mechanism underlies the considerable capacities in text generation. In this sense, LLMs can be considered as compressed and factorized Bayesian networks, where internal representations approximate latent variables modeling the underlying joint distribution (Wang et al., 2023a). The key to reasoning is the ability to generalize by identifying the right variables that capture the underlying structure of a task. Reasoning can thus be understood as approximating the latent variables of the underlying space: inference reduces to approximating the joint probability distribution over the subset of relevant variables, typically the Markov blanket in the underlying network. We summarize this perspective in Figure 1. Within our approach, deduction (Fagin et al., 2004), induction (Kyburg Jr & Teng, 2001; Schurz, 2019), and abduction (Douven, 2022) in LLMs all emerge as special cases of such approximation, described in Table 1.

Reasoning can operate on variables of different granularity. A basic form of reasoning operates on variables directly represented in token-level forward passes. In the setting of in-context learning (Dong et al., 2022), this corresponds to learning new tasks by implicitly encoding latent variables that serve as explanatory factors, which then condition the generation of appropriate propositions (Phan et al., 2023; Xie et al., 2021; Hahn & Goyal, 2023). Yet this approach has limits. Given the vast size of the underlying network, not all latent variables can be explicitly represented. Instead, models can dynamically construct additional latent

variables during inference, effectively extending beyond those encoded at training time, which underlies modern reasoning abilities in LLMs (Zhu et al., 2025; Chen et al., 2025a). Methods such as chain-of-thought prompting (Wei et al., 2022b) and zero-shot reasoning (Kojima et al., 2022) illustrate this principle: the model generates intermediate propositions that act as latent variables, which then condition subsequent steps of generation. This form of reasoning exhibits scaling laws, which can be interpreted as progressive refinements of the latent-variable space: greater compute and model capacity enable increasingly fine-grained approximations of the latent structure that supports reasoning (Snell et al., 2024; Geiping et al., 2025). Several approaches have been proposed to improve the generation and use of latent variables for reasoning. Prompting methods, such as Tree-of-Thought, or parallel sampling, can be interpreted within our framework as mechanisms for dynamically constructing additional latent variables during inference, which then further condition generation (Yao et al., 2023; Snell et al., 2024; Kamoi et al., 2024). Another line of work focuses on refining the latent space itself. Fine-tuning strategies, including reinforcement learning with human feedback (RLHF), adjust the latent structure to align reasoning with external objectives, thereby treating reasoning models as specialised adaptations of this general approach (Kumar et al., 2024). Other methods operate directly in latent space to enhance the quality of intermediate representations (Chen et al., 2025a). In other words, these approaches all serve to better approximate inference on an underlying, latent, Bayesian network of propositions. The objective is then to approximate as best as possible an ideal space of propositions under tractability constraints, and limit the failures of Bayesian reasoning (Chlon et al., 2025).

## 4 A Taxonomy of Reasoning Validity

The normative account of reasoning from Section 2 and its approximation in LLMs seen in Section 3 allow us to draw an account of valid and invalid reasoning in LLMs.

**Overview of the taxonomy.** The validity of reasoning depends on (i) the underlying Bayesian causal model of the joint probability distribution across propositions and (ii) the LLM's ability to perform correct inference over this structure. These two requirements jointly determine the conditions for success and the loci of failure. The immediate diagnosis of the failure of reasoning locates at the propositional level. Once this failure is understood at the propositional level, it is then possible to assess how it is instantiated at the token level, the limits of reasoning and devise potential solutions.

**Failures.** The components of the network yield a taxonomy of failure points and, conversely, suggest corresponding success criteria, summarized in Table 2. These failures happen regardless of the direction of inference on the Bayesian network. In all these cases, the forms of reasoning mentioned above (deductive, abductive, inductive, causal) would fail. This taxonomy provides a formal framework for explaining the failures of reasoning most frequently documented in the literature. These include errors in representing the underlying structure of reasoning processes (Wu et al., 2024), mis-specified formal logic (Parmar et al., 2024; Liu et al., 2025), failures in causal and counterfactual reasoning (Wang, 2024; Chen et al., 2025b), and systematic weaknesses in mathematical reasoning (Ahn et al., 2024; Yan et al., 2024), which can all be framed using the above formalization. Many failures of evidential reasoning can similarly be understood as arising from inadequate or misapplied conditioning. More broadly, critics have pointed to the lack of a stochastic or causal world model, which can be understood at the model layer of a taxonomy (Bender et al., 2021; Mitchell & Krakauer, 2023).

**Improvements.** In turn, these failures suggest a series of improvements and solutions. At the level of propositional abstraction, each failure identifies an area where reasoning can be strengthened. The resulting taxonomy of improvements helps to explain the substantial progress achieved in recent reasoning models. The most significant advances stem from expanding the variable scope and refining the topology and causal structure of the underlying network. Reasoning models generate more intermediate variables, coupled with more effective compression in latent space, which improves their reasoning capacities (Hao et al., 2024; Ruan et al., 2025). For example, specially tuned attention mechanisms have been proposed to sharpen edge representations, thereby enhancing the network's topology (Zelikman et al., 2025). Parameter-level improvements, which improve the estimation of conditional distributions throughout the network, have also been introduced (Wang et al., 2023b). Scaling laws can then be understood within this framework as

Table 2: Taxonomy of failure modes for LLMs representing reasoning over a Bayesian network.

| Category | Subtype | Failure | Description |
|----------|---------|---------|-------------|
| **Model layer (Bayesian network)** | | | |
| Structure | Variable scope & granularity | Variable set/state space mismatch: $V \neq V^*$ (e.g., latent $Z$ omitted; states too coarse) | Model omits or merges relevant factors to represent the underlying distribution |
| | Topology (edges) | Empirical vs. graph independences disagree: $\exists X, Y, Z : (X \perp Y \mid Z)_G$ but $(X \not\perp Y \mid Z)_{P_V^*}$ or vice versa | Model does not reflect the true links between variables |
| | Causal structure | $\hat{P}(Y \mid do(X))$ undefined or $\neq P_V^*(Y \mid do(X))$ | Model fails to encode correct interventional semantics. |
| Parameters | Priors | For root $X_i$ in $G$: $\hat{P}(X_i) \neq P_V^*(X_i)$ | Model miscalibrates marginals. |
| | Likelihoods | For any $X_i$: $\hat{P}(X_i \mid \mathrm{Pa}_G(X_i)) \neq P_V^*(X_i \mid \mathrm{Pa}_G(X_i))$ | Model's local conditionals do not match underlying data. |
| **Instance & Implementation (LLM)** | | | |
| Evidence | Incorrect conditioning set (noise) | Computed $P_\theta(Q \mid E')$ instead of desired $P_\theta(Q \mid E)$ | During inference, conditions on noise |
| | Incorrect conditioning set (propositions) | Computed $P_\theta(Q \mid E')$ instead of desired $P_\theta(Q \mid E)$ | During inference, conditions on the wrong evidence set. |
| | Selection bias | Conditioning on collider/descendant $C$: $X \perp Y$ in $G$ but $X \not\perp Y \mid C$ | Opens spurious dependence; biased estimates. |
| Inference | Approximate posterior error | $q(Q \mid E) \neq \hat{P}(Q \mid E)$ (bias, variance, non-convergence) | Inference not consistent with probability rules. |
| | Incorrect model | Inference run on $\tilde{P} \neq \hat{P}$ (wrong factors) | The wrong model or Markov Blanket is used for inference. |
| | Incorrect causal reasoning | Used $\hat{P}(Y \mid X)$ to answer $\hat{P}(Y \mid do(X))$ | Observational conditioning substituted for intervention. |

*Notation.* $P^*$: true distribution over the true variables $V^*$. $\hat{P}$: model's estimated joint distribution over $V$, induced by the Bayesian network $(G, \theta)$. $P_V^*$: marginal of $P^*$ onto the model's variables $V$. The Bayesian Network uses variables $V$, graph $G$, parameters $\theta$, inducing $P_\theta$. $\mathrm{Pa}_G(X_i)$: parents of $X_i$ in $G$. $Q$: query variables. $E$ (desired) and $E'$ (used): evidence assignments on $V$. $q(\cdot)$: approximate posterior. $do(\cdot)$: intervention operator. $X \perp Y \mid Z$ / $X \not\perp Y \mid Z$: (conditional) independence/dependence.

progressively refining latent-variable spaces while maintaining compression (Wu et al., 2025; Geiping et al., 2025). These approaches can be seen as direct attempts to improve the internal representation of the Bayesian model within the latent space of language models.

# 5 Challenges and Prospective Solutions

Although Table 3 provides a framework for identifying both the failures and the potential solutions for language model reasoning, its implementation at scale remains challenging.

## 5.1 Challenges

The central difficulty lies in bridging levels of abstraction: reasoning operates at the level of propositions, while language models are trained in token space. This creates a semantic gap, as there is no guarantee that optimization in token space will converge to correct reasoning at the propositional level. The training objective is next-token prediction, not reasoning itself: it maximises likelihood under observed data at the token level. This mismatch introduces noise and limitations, such as spurious correlations, sensitivity to token-level patterns, and failures under distribution shift. This restricts, by its very nature, how reliably

Table 3: Solutions aligned with the failure taxonomy in Table 2. The Model level lists solutions at the level of abstraction of the propositional space and then assesses how to implement these solutions at the language model level.

| Category | Subtype | Model Level (Bayesian Net) | LLM Level |
|---|---|---|---|
| **Model layer (Bayesian network)** | | | |
| Structure | Variable scope & granularity | Augment $V$ with latent $Z$; refine state partitions; fit to $P_V^*$ | Better representation learning of latent variable encoding. It is a decision boundary problem. |
| | Topology (edges) | Learn $G$ via score+CI tests; orient with interventions | Attention needs to reflect the link between propositions. |
| | Causal structure | Identify $P(Y \mid do(X))$ (ID/back-door/front-door); implement mutilated graph; sensitivity to latent $Z$ | Disambiguate the latent representations. The causal effect needs to be modeled. |
| Parameters | Priors | Set roots to $P_V^*(X_i)$ (empirical Bayes) | Generation does not reflect underlying proposition priors. Incorrect generalisation. |
| | Likelihoods | Refit $\hat{P}(X_i \mid \mathrm{Pa}_G(X_i))$ | Generation does not reflect underlying proposition distribution. Incorrect generalisation. |
| **Instance & Implementation (LLM)** | | | |
| Evidence | Incorrect conditioning (noise) | Canonicalize $E$ to ensure it is a valid proposition | Conditioning on noise, such as irrelevant tokens |
| | Incorrect conditioning (propositions) | Conditions on colliders such as text | Conditioning on the wrong propositions |
| | Selection bias | Conditioning on collider/descendant $C$: $X \perp Y$ in $G$ but $X \not\perp Y \mid C$ | Conditioning on spurious dependence and biased estimates. |
| Inference | Approx. posterior error | Exact elimination when tractable; else approximation via MCMC or similar methods | Partial reasoning, does not account for all paths |
| | Incorrect model used | Factor registry with scope | Incorrect latent representations used for modeling the query. |
| | Incorrect causal reasoning | Separate estimators for $P(Y \mid X)$ and $P(Y \mid do(X))$ | Disentanglement of which latent variables are causal and which observational. |

the learned distribution can support inference over propositions. As LLMs approximate the distribution of tokens, this does not ensure alignment with propositional reasoning. The space of propositions is vastly larger than training on the fixed vocabulary $\Sigma$, and thus training on it is computationally intractable. A challenge underlying the above failures ensuring that propositions can be approximated and made tractable within token-based representations, without disrupting learning dynamics. Overcoming this challenge means learning the underlying joint distribution over propositions directly into the model's representations, with the degree of compression determined by architectural and training constraints. Given the scale of this space, such structures will need to be learned in an emergent and scalable way, arising from the training procedure itself. The solution must therefore be (1) general across propositions and (2) tractable. In this way, we reframe the challenge of reasoning as a specific category of the alignment problem, which we term *epistemic alignment* (Gabriel, 2020; Anwar et al., 2024). By this, we mean orienting models toward an epistemic normative framework that serves as a reference point.

## 5.2 Prospective Solutions

This section offers prospective solutions for overcoming the above challenge, with theoretical and empirical improvements.

**Foundations of reasoning.** Progress in language model reasoning has been largely empirical; a formal account that maps the propositional space onto the internal geometry of LLMs is still missing. Such an account would provide a framework for further experimental work, in turn helping to improve the theory. We sketch a program along four key questions, mapping to the core part of our reasoning framework:

*(i) Representation.* The foundational element of reasoning would be the representation, and how the latent representations are able to represent propositions and reflect the joint distribution in an efficient way. This implies an understanding of how these propositions are stored in latent spaces, and the structure and nature of the compression. It also implies understanding the connection of features in circuits (Tigges et al., 2024). A second step is to make these latent representations interpretable: decompose the propositional latent space into identifiable components with stable semantics (variables, states, and their relations). Token-level mechanistic interpretability provides useful tools at the token level (e.g., interpretable feature extraction), which could then be extended at the propositional level (Bereska & Gavves, 2024; Ameisen et al., 2025).

*(ii) Emergence and training.* A further aspect of research would study the emergence of these propositional latent representations under next-token training. Such a theory would seek statistical conditions (data, scale, architecture) under which propositional structure appears and stabilises; and bounds that relate compression to fidelity. This connects to the statistical origins of emergent structure as manifold learning, which would in turn lead to the equivalent of scaling laws for propositional representations (Whiteley et al., 2022; Wei et al., 2022a; Modell et al., 2025; Watanabe, 2009; Amari, 2016; Ay et al., 2017; Geiping et al., 2025).

*(iii) Inference.* Given an encoding of propositions, a theory of reasoning must specify how beliefs are represented and updated and how propositions are combined during multi-step computation in latent space. We view inference as approximate posterior updating over a compressed model of propositions. An initial step would be to identify the operators on the propositional latents. This would imply identifying the posterior object and an update operator. Given the geometry of the internal space, this can also be understood geometrically as information-geometric projections on the learned manifold (Watanabe, 2009; Amari, 2016; Ay et al., 2017). The mapping between the latent belief update and the underlying propositional space would provide a clear understanding of the underlying reasoning process. Once the representation and update operators are fixed, a theory of correct reasoning at the latent level should specify when inference remains on–manifold and when it fails. Concretely, it should characterise conditions under which messages keep the posterior inside the support of the learned distribution, and when they drive it off–manifold (Kumar et al., 2025). A last area of research would be reasoning interpretability. If the propositional latent space is interpretable, intermediate updates can be audited step-by-step for validity, providing explainability and control as systems scale.

*(iv) Evaluation.* Evaluating reasoning at scale remains open: the propositional space is vast, and reasoning unfolds over long trajectories. An adequate approach must (a) distinguish coverage from accuracy and make explicit which subspace of propositions and which classes of reasoning chains are assessed, as well as (b) quantify local error and partial alignment rather than reporting a single aggregate score (Shen et al., 2023; Cao et al., 2024). In practice, models may omit relevant variables or represent them only approximately, which must be evaluated to avoid emergent safety risks (Anwar et al., 2024).

**Architecture.** A central question is what model architecture is most expressive for capturing the underlying Bayesian graph without introducing systematic errors. One direction is to enforce structural constraints directly within the architecture so that variables and their topological relations are better represented.

*(i) Representational Space.* Architecture can be adapted to better approximate the propositional latent. A first step has been to perform reasoning directly in the latent space (Chen et al., 2025a; Ruan et al., 2025; Zhu et al., 2025). The issue is that these latent spaces are not fully aligned to propositional distributions and would need further improvements. Given the shape of the propositional manifold, geometric constraints can be applied to ensure that the learned propositional manifold better approximates the underlying propositional

data (Amari, 2016; Modell et al., 2025). A second approach would be to enforce constraints to disentangle the token space from the propositional space, which involves solving a specific case of superposition (Hänni et al., 2024; Elhage et al., 2022).

*(ii) Information Flow.* Architecture can also ensure inference and information flow. A key component of modern transformer architecture is the attention mechanism, which is responsible for conditioning on prior text. Better reasoning would imply better adapted attention mechanisms and architecture, able to condition on propositions across the context window (Wiegreffe & Pinter, 2019). Recent progress in reasoning, for instance multi-head latent attention, points towards the attention mechanism as a key component of the approximation of the propositional structure.

*(iii) Causal.* Encoding constraints in the representations would help disambiguate counterfactual structure, for instance by constraining the attention patterns to better approximate the underlying structure (Melnychuk et al., 2022) This could find the encoded causal graph structure within the LLM's internal space, and separate it from other processes (Brahma et al., 2015; O'Neill et al., 2024).

**Training.** The goal of training would be to approximate the underlying propositional graph within the model's representations. This would imply data that adequately captures the true joint distribution of propositions, and objectives that can learn this pattern efficiently.

*(i) Data.* A first step would be to train with token-level datasets with higher propositional content, for example, curriculum learning covering various graph pathways to reduce entanglement (Kumar et al., 2025; O'Neill et al., 2024). A second step would be to train with proposition-level datasets, which would range from contrastive pairs to propositional traces, structured rationales (team et al., 2024).

*(ii) Objectives and algorithms.* Reinforcement learning could be used with proposition-level rewards, as well as reinforcement learning on explicit or constructed propositional graphs (Rafailov et al., 2024; Kumar et al., 2024; Matsuo et al., 2022). Objectives that shape the propositional structure could be used, such as contrastive learning over proposition on the intermediate states (Reimers & Gurevych, 2019).

**Inference.** On the experimental side, the goal is to exploit the learned propositional space as effectively as possible, improving fidelity and computational efficiency. Holding the model fixed, we can improve reasoning by (a) specifying the query and evidence precisely, and (b) tightening the approximation to the Bayesian update during decoding.

*(i) Prompting and verification.* Improved prompting can exploit the structure of the latent space to better approximate reasoning. Structure reasoning chains (e.g., tree/beam exploration, self-consistency) can aggregate over multiple candidate posteriors and stabilise answers (Snell et al., 2024). Self-correction, reflection, and verifier/checker loops refine intermediate propositions and reject inconsistent steps (Welleck et al., 2023; Kamoi et al., 2024).

*(ii) Inference scaling.* Improving inference scaling can make reasoning cheaper and faster. Empirical approaches can identify techniques to reason at scale (Wu et al., 2025; Snell et al., 2024; Muennighoff et al., 2025).

*(iii) Adaptability.* Efficient reasoning requires dynamically allocating computational resources across steps and subproblems, aligning the resource usage with the expected value of computation (Griffiths et al., 2015; Lieder & Griffiths, 2020). Practically, this would entail experiments on dynamic branching of queries and different reasoning paths based on the scale.

## 6   Discussion and Conclusion

We have proposed a unifying framework for reasoning in LLMs: propositions as the basic objects, probabilistic inference as the update rule, and a compressed Bayesian network as the abstract structure that the model ought to approximate. Framed in this way, progress in reasoning becomes a problem of epistemic alignment: making token-level training and architectures converge to proposition-level correctness under

tractable approximations. Although we have only sketched the core ideas, this framework provides a base for several further use cases:

1. **Temporal reasoning.** Extend the Bayesian formulation to discrete time (dynamic Bayesian networks or temporal structural causal models). This yields time-indexed propositions and provides a normative account of reasoning across time (Pearl, 2014; Russell & Norvig, 2022).

2. **Deontic reasoning.** Deontic reasoning can be framed as a further extension of the framework, where reasoning chains could follow deontic logic (Von Wright, 1951). Consequentialist views could enter via utility over outcomes and deontological views with obligations encoded as constraints on reasoning (Rao et al., 2023; Gabriel, 2020).

3. **Multimodal reasoning.** Propositions can be grounded in a world model spanning language, perception, and action. Cross-modal latents could link textual propositions to visual or sensorimotor variables, enabling reasoning across modalities with the same Bayesian infrastructure (Patel & Pavlick, 2022; Yan et al., 2024).

4. **Agentic reasoning.** Bayesian inference can naturally be extended with choice via Bayesian decision theory. This would define further utilities, policies, and risks over the propositional graph, which in turn would enable a direct coupling between action and reasoning (Russell & Norvig, 2022).

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
