# OpenReview forum: "A Normative Framework for Reasoning in Language Models"
_TMLR — Withdrawn by Authors_

### Review · Reviewer_538P · 2025-12-31

**Summary Of Contributions:**

This paper addresses the theoretical gap in understanding how LLMs reason. The authors propose a normative framework that defines reasoning as probabilistic inference over propositions.  The paper argues that while LLMs are trained on next-token prediction, they implicitly induce an approximate distribution over propositional content. Consequently, LLMs can be viewed as compressed or factorized Bayesian networks where internal representations serve as latent variables approximating the true joint distribution of propositions. Based on this theoretical mapping, the paper contributes:

1. A structured classification of errors divided into the "Model Layer" (failures in the underlying Bayesian network structure, topology, or parameters) and the "Instance/Implementation Layer" (failures in evidence selection, conditioning, or approximate inference)
2. A proposal to align token-level training objectives with proposition-level normative standards
3. Prospective solutions spanning architecture, training, and inference to bridge the semantic gap between tokens and propositions.

**Key Strengths**:
1. The framing of “propositions + Bayesian inference” provides a coherent umbrella to discuss disparate reasoning modes—deductive, inductive, abductive, and causal—and connect them to a single normative standard. This effectively synthesizes various phenomena (like chain-of-thought and in-context learning) under one theoretical roof
2. The structured breakdown of failures in Table 2  is a strong diagnostic tool. Distinguishing between errors in the world model (e.g., variable scope, causal structure ) versus errors in the inference process (e.g., conditioning on noise, selection bias ) offers a more precise vocabulary than generic terms like "hallucination.
3. The paper explicitly highlights the "semantic gap" between token-level training and proposition-level correctness. This motivation for "epistemic alignment" is compelling and well articulated.

**Key Weaknesses**:

1. The connection between math and text sounds vague. The paper's core argument relies on translating "token strings" (text) into "propositions" (facts) and then into a "Bayesian Network" (a math model). However, it never explains how to actually do this in practice.For example, the paper defines the probability of a proposition as the total "probability mass" of all strings that express it. But it does not explain how to determine if two different sentences (e.g., "The sky is blue" vs. "The color of the sky is blue") count as the same proposition. Without a clear rule for grouping these sentences, the theory is hard to test.

2. Limited novelty: Much of the "new" framework is actually a collection of ideas from other papers.For instance, the idea that LLMs are "latent variable models" and that in-context learning is "implicit Bayesian inference"  are already established concepts. The paper’s main contribution is organizing these ideas into a list of failures (the taxonomy), but it presents the background theory as if it were a new discovery. It should be clearer that this is a synthesis of prior work.

3. Need more empirical evidence: The paper is entirely theoretical. It does not provide a single experiment or concrete example to prove its points.It would be much stronger if it took one specific reasoning problem, showed how an LLM fails at it, and then demonstrated exactly how that failure fits into their "Table 2" taxonomy. As it stands, the reader has to trust the theory without seeing it in action.

4. The paper makes very strong claims, such as stating that Bayes' rule is the only rule that "guarantees coherence of belief". This ignores other established frameworks for uncertain evidence, such as Jeffrey Conditioning. Such unqualified statements weaken the paper's theoretical precision.

**Audience:**

Yes

**Audience Explanation:**

I think the TMLR audience will be interested to read the findings of this paper. The "reasoning vs. retrieval" debate is a central pillar of current LLM research. As models like DeepSeek-R1 and O1 increase test-time compute, the community needs a formal language to describe what is happening during these "thinking" steps. This paper provides that language. Researchers in alignment, interpretability, and cognitive modeling will find the taxonomy of failure modes (Table 2) and the proposed solutions (Table 3)  highly relevant for designing more robust benchmarks and architectures.

**Broader Impact Concerns:**

I do not see major direct ethical concerns specific to the proposed framework. The paper is a theoretical framework focused on improving the accuracy and interpretability of reasoning. If successful, it would lead to more reliable and transparent AI systems, which generally aligns with safety goals. The authors already touch upon "epistemic alignment", which addresses potential risks associated with misaligned reasoning processes.

**Claims And Evidence:**

No

**Claims Explanation:**

The paper is largely a conceptual/normative proposal, so most support comes from definitions and high-level argument rather than empirical validation.
 While the Bayesian/propositional framing and the failure-mode taxonomy are clearly motivated and internally coherent, several core claims are not backed by convincing, operational evidence:

1. The characterization of LLMs as compressed/factorized Bayesian networks reads partly as metaphor/interpretation rather than a testable claim, and the paper does not provide empirical or formal validation of this mapping.
2. Some normative statements (e.g., about the uniqueness of Bayes as an update rule) would benefit from clearer assumptions/qualification to be fully accurate.
3. The taxonomy is useful, but without case studies or quantitative evaluation, it’s not demonstrated that it explains real LLM errors better than existing accounts.
4. The bridge from token-level distributions to proposition-level inference (strings → propositions → probabilities over propositions) is asserted at a high level but not operationalized, so the “success criteria” and some diagnostics are hard to verify in practice.

**Requested Changes:**

**Critical for Acceptance**:

1. Operationalize the Proposition Mapping (Section 3): The current definition of a proposition's probability ("total probability mass... of strings that express that proposition") is theoretically sound but practically undefined. You must provide an operational definition of proposition identity. Specifically, how does the framework handle paraphrases, entailment, or underspecification?

2. Provide a Grounded Case Study: The paper is currently entirely abstract. To demonstrate that the failure taxonomy in Table 2  is a valid diagnostic tool rather than just a theoretical hypothesis, you must ground it in a specific example. Please include a worked case study (even a synthetic one) that: (a) defines a simple reasoning task and its ideal Bayesian Network; (b) identifies a specific LLM error; and (c) explicitly maps that error to a category in your taxonomy (e.g., distinguishing a "Topology" error from a "Parameter" error).

3. The text oscillates between treating the "LLM-as-Bayesian-Network" as a useful interpretive metaphor and as a literal description of the model’s internal mechanism.

**Strengthening the Work**:

1. The claim that Bayes' rule is the unique coherence-preserving update rule is too strong and ignores established frameworks for uncertain evidence, such as Jeffrey Conditioning.Qualifying this statement to acknowledge soft updates or uncertain evidence would improve the theoretical precision of Section 2.

2. The taxonomy mentions failures like "Empirical vs. graph independences disagree."Briefly outline how a researcher would actually test for this. What statistical test or intervention on the LLM's activations would reveal this specific disagreement? Adding this would make the taxonomy significantly more actionable for the community.

3. Expand on Scaling Laws (Section 3) The paper links scaling laws to "progressive refinements of the latent-variable space". It would strengthen the paper to discuss if there is existing empirical evidence showing that larger models actually represent a more complete Markov blanket, or if this is purely a hypothesis.

4. Architectural Constraints: Briefly discuss how current "mixture-of-experts" or "sparse attention" mechanisms might specifically hinder or help the formation of the "Learned Network" described in Figure 1. This is interesting to learn how newer architectures adapt to this theory.

---

### Review · Reviewer_3hvS · 2026-01-06

**Summary Of Contributions:**

The work presents a normative framework for viewing LLM reasoning as probabilisitic inference over logical propositions. They start by representing reasoning with Bayesian Networks and relate this to how an LLM reasons over an approximate network over latent variables. They then outline failure modes of reasoning in Bayesian Networks alone and as viewed as LLMs. For each failure mode, outline possible solutions to address them. They provide a general roadmap for performing "epistemic alignment" which aims to bridge next-token prediction with more accurate reasoning at the proposition level.

Strengths:
- Sound formal connection between a Bayesian Network over propositional variables and LLM reasoning.
- Clear description of failure modes of reasoning and associated solutions in Bayesian Networks

Weaknesses
- Some reasoning failure modes in the "Instance & Implementation" section of Table 2 are lacking details and explanation
- Solutions in the "Instance & Implementation" section of Table 2 are vague, lacking explanation, and do not outline how to implement the solutions as stated in the table caption.
- The "semantic gap" between tokens and propositions requires more analysis and evidence.

**Audience:**

Yes

**Audience Explanation:**

Yes, the framing of LLM reasoning as Bayesian networks over propositional variables can incite further theoretical investigation. The roadmap provided for closing the semantic gap between tokens and propositions might be an interesting starting point for further research in this direction.

**Claims And Evidence:**

No

**Claims Explanation:**

No, the paper claims to provide a taxonomy of failures in LLM reasoning under the normative framework proposed, however the taxonomy is unclear and lacking in detail, and the proposed solutions are vague and not actionable.

**Requested Changes:**

- Fix citation after "Probabilistic reasoning extends this framework to uncertainty over propositions"
- In Table 2 under "Instance & Implementation (LLM)", can you make the distinction between "Incorrect conditioning set (noise)" and "Incorrect conditioning set (propositions)" more clear, perhaps notationally? Currently, their failures are the same. Also, it is unclear what the noise is in this case, should be rigourously defined.
- It is not clear to me what the difference between $q(Q | E)$ and $\hat{P}(Q | E)$. Could you include further explanation of $q$, or point me to where it is explained.
- What is $\tilde{P}$? Needs more explanation.
- It would be helpful if for each failure mode there was some kind of example, otherwise it is unclear if these failures are relevant in practice.
- In Table 3, all the solutions presented in the LLM level column of the Model Layer section are vague. For example, in "Better representation learning of latent variable encoding. It is a decision boundary problem.", can you explicitly define what the decision boundary problem is? Furthermore, for the Parameters category, the presented solution does not appear to be an actionable solution.
- In the Instance & Implementation section of Table 3, the solutions in the model level column require more explanation and most of the solutions in the LLM level column are not actionable towards implementation as claimed in the table caption.
- Regarding the semantic gap between tokens and propositions, you state "This mismatch introduces noise and limitations, such as spurious correlations, sensitivity to token-level patterns, and failures under distribution shift" but there is no evidence or explanation to how these arise. Perhaps a more rigorous definition of the semantic gap could be a start.
- I think the paper would greatly benefit from breaking down Table 2 and 3 into thorough explanations as they should highlight the importance of the proposed normative framework but as of now fail to do so due to lack of clarity and details.

---

### Review · Reviewer_pmUB · 2026-01-07

**Summary Of Contributions:**

The submission proposes a normative account of reasoning in LLMs that treats reasoning as probabilistic inference over propositions, using Bayesian networks as the reference abstraction. It then (i) gives a typology of reasoning modes (deductive/inductive/abductive/causal), (ii) derives a taxonomy of reasoning failure modes split into “model-layer” vs “instance/inference-time” errors, and (iii) maps these failures to high-level remedies plus a forward-looking research roadmap for improving proposition-level correctness under tractability constraints.

Key strengths
1. Clear organizing lens: Using Bayesian networks/probabilistic inference to unify disparate “reasoning” discussions provides a coherent conceptual scaffold.
2. Useful failure decomposition: The failure taxonomy (model structure/topology/causality/parameters vs evidence/selection bias/inference errors) is a helpful checklist for diagnosing where reasoning breaks.
3. Bridges causal and statistical reasoning: Explicitly surfacing causal vs observational reasoning and selection bias as failure modes is valuable, and often missing in LLM discussions.

Key weaknesses
1. Core mapping is under-specified: The bridge from token distributions to “propositional variables” and a Bayesian network is mostly asserted/illustrated, not concretely instantiated (e.g. what counts as a proposition, how equivalence across paraphrases is handled, how variables/edges are identified).
2. Claims outpace support: Some strong statements (e.g. about “necessary” improvement of propositional modeling from next-token modeling; uniqueness/coherence statements) are not carefully scoped or supported with formal argumentation or evidence.
3. Primarily a position/taxonomy piece: There is little in the way of falsifiable predictions, experiments, or an operational evaluation protocol that would let readers test the framework’s claims and usefulness.

**Audience:**

Yes

**Audience Explanation:**

Even though the work is largely conceptual, I expect some researchers working on LLM reasoning, evaluation, interpretability, and causal reasoning would find value in:
- The BN/propositional framing as a common vocabulary, and
- The failure-mode taxonomy as a diagnostic checklist and a way to structure evaluations/interventions.

So I do think it clears the “interest” bar for a subset of TMLR readers, even if the ultimate impact depends on tightening the claim-evidence connection.

**Broader Impact Concerns:**

None.

**Claims And Evidence:**

No

**Claims Explanation:**

The paper provides definitions, conceptual arguments, and taxonomies, but the strongest “bridging” claims are not supported with sufficiently concrete evidence or formalization to be convincing:

The submission says it “shows how this abstraction can be instantiated in LLMs” and “formalise success criteria for proposition-level correctness,” but the instantiation remains mostly conceptual (illustrations/tables) rather than an explicit mapping, worked example, or testable procedure.

Several key assertions read as broad or unconditional (e.g. about what next-token likelihood improvements imply for proposition-level correctness, and about normative uniqueness statements) without proof, careful conditions, or counterexample discussion.

The proposed “solutions” are largely high-level correspondences rather than implementable methods or evidence-backed prescriptions. As a result, the “prescriptive” claim is not fully earned.

Given TMLR’s emphasis that any claim-evidence gap should be addressed either by more evidence or reduced claims, this currently lands on the “gap remains” side.

**Requested Changes:**

Critical for acceptance (in my view):

1. Either add a concrete instantiation + evidence, or explicitly reduce/reframe claims.
- Add at least one worked example (toy domain is fine) that defines: propositions, the mapping from text to propositions/variables, how probabilities over propositions are derived, and how 2-3 failure modes manifest and are detected.
- Alternatively, reframe explicitly as a position/survey/taxonomy paper and tone down claims about “formalised success criteria,” “showing instantiation,” and being “prescriptive.”

2. Make the proposition-level semantics operational.
Specify what counts as a proposition in practice, how paraphrase/equivalence is treated, how context dependence is handled, and what “proposition-level correctness” means in measurable terms.

3. Tighten or justify strong normative/technical claims.
For statements like “X necessarily implies Y” or “Bayes is the unique coherence-preserving update,” either:
- Add formal conditions and citations that make the claim accurate, or
- Soften to scoped claims that match what’s actually established in the paper.


Would strengthen the work (but not strictly required):

5. Add an explicit evaluation protocol aligned to the taxonomy.
E.g. propose benchmark design patterns or metrics that separately test: variable scope mismatch vs inference-time conditioning errors vs causal/selection bias errors.

6. Clarify novelty and relationship to closely related viewpoints.
More sharply distinguish what is new here versus prior work that frames LMs as latent-variable/Bayesian(-in-expectation) models, and discuss limitations of the BN analogy.

7. Improve “solutions” by making at least one remedy actionable.
Pick one failure class and outline a concrete LLM-side intervention (training objective tweak, inference-time method, or diagnostic test) that follows from the framework.

---

### Note · Authors · 2026-03-06

**Comment:**

Dear editor and reviewers,

Thank you for your kind and helpful feedback. This has helped us considerably improve the paper over the past weeks.

We especially appreciate the focus on an operative diagnostic, which revealed a limitation in our approach. Devising a tractable algorithm for this diagnostic required us to develop new abstractions that have since reframed the direction of the work.

We have therefore decided to withdraw the paper, as completing these revisions to the standard we feel is necessary will take us longer than the revision period granted. We intend to resubmit once the work is in a stronger state.

We are grateful for the time and effort the reviewers and editors who helped improve this work.

Best regards,
Matthieu on behalf of the Authors

**Withdrawal Confirmation:**

I have read and agree with the venue's withdrawal policy on behalf of myself and my co-authors.